# Catalytic stereodivergent allylic alkylation of 2-acylimidazoles for natural product synthesis

Ruimin Lu[1,2], Qinglin Zhang[1,2] & Chang Guo [1] ✉

The stereocontrolled allylic alkylation of carbonyl compounds with the goal of producing the full range of stereoisomers presents an effective approach for increasing the productivity of collective natural product synthesis and the creation of chiral molecule libraries for drug exploration. The simultaneous control of regio-, diastereo-, and enantioselectivity poses a significant synthetic challenge in contemporary organic synthesis. Herein, we describe a catalytic stereodivergent α-allylation protocol applicable to both aliphatic and aromatic 2-acylimidazoles, thereby providing a practical blueprint for the divergent synthesis of important chiral building blocks. Each of the six isomeric α-allylated compounds can be readily obtained with remarkable yields and exceptional stereoselectivities, by judiciously selecting the appropriate leaving group and permutations of enantiomers adapted from nickel and iridium catalysts. The versatility of this asymmetric allylic alkylation has been successfully utilized in the enantioselective synthesis of (*R*)-arundic acid and (*S*,*S*)-cinamomumolide, as well as in the stereodivergent total synthesis of tapentadol.

The configuration of stereocenters in organic compounds is critical in determining the intricate functionalities exhibited by vital biomacromolecules, including proteins and nucleic acids, as well as a wide range of biological compounds (Fig. 1a, e.g. arundic acid, mitiglinide, SDZ 242-484, cinamomumolide, tapentadol, and paroxetine)[1]. Consequently, a comprehensive understanding of the stereochemical structure-activity relationships necessitates the exploration of all potential stereoisomeric variations inherent to a given natural product or lead candidate[2]. The simultaneous and controlled creation of multiple stereogenic centers in a single catalytic step, ensuring both the relative and absolute configurations, remains an essential area of research[3]. Specifically, the enantioselective α-allylic alkylation represents a potent and straightforward approach within chemical research for the construction of carbon−carbon bonds with high enantioselectivity[4]. In this context, α-allylated tertiary carbonyls serve as pivotal synthetic intermediates for the synthesis of structurally diverse pharmaceuticals and natural products.

Synergistic dual catalysis has emerged as a compelling strategy for efficiently driving contemporary synthetic processes[5–11]. Carreira and co-workers introduced the concept of stereodivergent dual catalysis[12], wherein two distinct catalysts synergistically engage in orthogonal catalytic functions enabling the controlled formation of two new stereogenic centers (Fig. 1b)[13–23]. Remarkable advancements have been made in this domain through the synergistic interplay of discrete catalytic processes to achieve stereodivergent product formation[24–33]. Various nucleophilic substrates, including α-hydroxy ketones[34], aldimine esters[35–37], pentafluorophenyl esters[38–40], azaaryl acetamides[41–43], and oxindoles[44] have been successfully employed in these dual-catalyzed stereodivergent synthesis. The ability to access all possible stereoisomers of both linear and branched α-allylated tertiary carbonyls through catalytic stereodivergence presents a fascinating avenue in terms of atom and step economy (Fig. 1c)[45–55]. Recently, Zheng, Wu and coworkers reported a seminal Ni/Pd dual-catalysis for the asymmetric alkylation of 2-acylimidazoles to generate linear

---

[1]Hefei National Research Center for Physical Sciences at the Microscale and Department of Chemistry, University of Science and Technology of China, Hefei 230026, China. [2]These authors contributed equally: Ruimin Lu, Qinglin Zhang. ✉e-mail: guochang@ustc.edu.cn

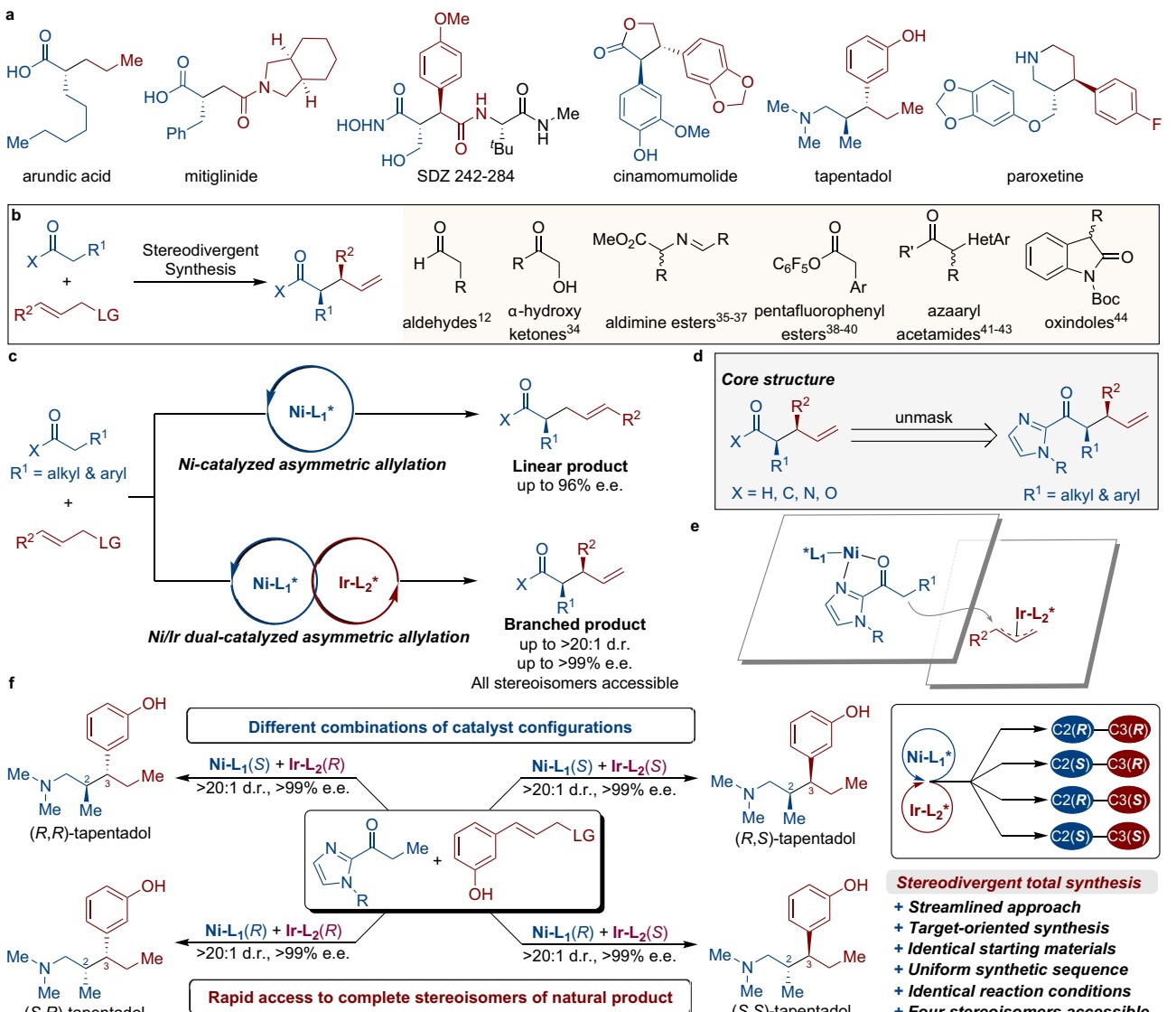

**Fig. 1 | Overview of regio- and stereodivergent α-allylation of 2-acylimidazoles.**
**a** Representative chiral biologically relevant compounds. **b** Prior work: different carbonyl-pronucleophiles used in stereodivergent catalysis involving π-allylmetal electrophiles. **c** This work: catalytic stereodivergent access to all possible regio- and stereoisomers of both linear and branched α-allylated tertiary carbonyls with excellent optical purity. **d** Easy post-synthetic transformation of acyl-imidazoles into various carbonyl derivatives. **e** Ni/Ir dual-catalyzed stereodivergent α-allylation. **f** Target-oriented stereodivergent total synthesis of tapentadol via Ni/Ir dual catalysis.

allylated products[56]. Despite extensive research efforts, the constrained applicability of conventional alkyl-substituted carboxylic acid derivatives as nucleophilic reagents within the realm of catalytic stereodivergent synthesis, primarily due to the inherent intricacies associated with enolization, impedes the exhaustive exploration of all potential isomeric configurations exhibited by structurally diverse natural products[57–59].

In continuation of our previous work concerning dual catalysis for achieving stereodivergent total synthesis[59], we postulated that a synergistic dual catalyst system comprising a chiral Lewis acid and a chiral iridium complex would present a highly efficient strategy for the stereodivergent α-allylations of 2-acylimidazoles. Several potential advantages of this approach warrant highlighting: (1) the bidentate coordination of 2-acylimidazoles with a chiral Lewis acid enhances the nucleophilicity of the carbon atom while exhibiting commendable stereocontrol (Fig. 1d)[60, 61]; (2) the configurations of the two contiguous tertiary carbon stereocenters in the resulting product, each originating from the same starting 2-acylimidazole and allyl species, can be controlled by the chiral Lewis acid in conjunction with the chiral

iridium complex (Fig. 1e). Consequently, manipulating the configuration of the two chiral catalysts will readily afford the four stereochemically distinct isomers of the branched products[8]; (3) direct synthetic approaches afford impressive enantioselectivities and allow precise regulation of the regioselectivity (branched versus linear products), effectively facilitating the creation of the complete array of isomers in a divergent manner; and (4) facile post-functionalization of the acylimidazole to yield common carbonyl analogs (e.g. aldehydes, ketones, esters, amides, alcohols, etc.) can be achieved. Remarkably, this post-functionalization can be accomplished while simultaneously preserving the enantioselectivity[61], thereby facilitating collective stereodivergent total synthesis.

Herein, we present a highly selective approach for regio- and stereodivergent allylic alkylation reactions, facilitating the synthesis of diverse α-allylated acylimidazoles with exceptional regio-, diastereo-, and enantioselectivity. The simplicity and efficiency of this approach open up avenues for the enantioselective synthesis of (R)-arundic acid and (S,S)-cinamomumolide, as well as the total synthesis of all four tapentadol stereoisomers (Fig. 1f).

## Results

### Optimization studies

Our preliminary investigations into the divergent synthesis of linear and branched α-allylated acylimidazoles began with the utilization of 2-acyl imidazole (**1a**) as the nucleophile and allylic derivatives (**2**) as the electrophile (Table 1). In the presence of 10 mol% of nickel acetate and the chiral diamine ligand (*S,S*)-**5a**, the linear allyl substitution reaction of **1a** and **2a** occurred with 22% yield and 28% e.e. (entry 1). Other chiral ligands were subsequently examined (entries 2 and 3), and the use of (*S,S*)-**5c** resulted in the enantioenriched synthesis of linear product **3a** in 80% yield and excellent e.e. (entry 3)[56]. Having established an asymmetric methodology for the synthesis of linear homoallylic 2-acylimidazoles, we attempted to address the synthesis of branched regioisomers with precise control over stereoselectivity. The combination of 2 mol% of iridium catalyst [Ir(COD)Cl]$_2$/(*S,S*)-**6** with nickel catalyst led to a regioselectivity switch in the allylation reaction, and the branched product **4a** was obtained in good stereoselectivity (entry 4). Subsequently, the evaluation of the leaving group (LG) of the allylic substrates (entries 5 and 6) led to the identification of the optimal conditions using substrate **2c**, providing **4a** in 85% yield, >20:1 d.r., and >99% e.e. (entry 6). Control experiments verified that both metal catalysts remained inert in the absence of additional ligands (entries 7 and 8), and further investigations confirmed the indispensability of each component within the integrated catalytic system (entries 9 and 10).

### Substrate scope

We extended the application of our developed methodology to enable the selective synthesis of six distinct isomers of the α-allylated product in a divergent manner (Fig. 2). Initially, we explored the asymmetric allylation of 2-acyl imidazole (**1a**) with cinnamyl bromide (**2a**) via nickel catalysis, and both enantiomers of **3a** could be achieved in excellent results through the utilization of chiral ligands with distinct configurations (Fig. 2a). To assess the stereodivergence potential of the cooperative Ni/Ir-catalytic system, we carried out the branched-selective alkylation reactions of 2-acyl imidazole (**1a**) with cinnamyl phenyl carbonate (**2c**) (Fig. 2b). Notably, all four possible stereoisomers of **4a** were successfully obtained with comparable efficiency, with both stereogenic centers being fully controlled under the independent guidance of the nickel and iridium catalysts, respectively.

With the optimized conditions for linear allylic alkylation (Table 1, entry 3), we explored the scope with respect to the 2-acyl imidazoles (**1**) and allylic bromide derivatives (**2**). As summarized in Fig. 3, diverse alkyl substrates (**1a**–**1g**) were successfully applied to this reaction efficiently and gave the desired products in great yields (**3a**–**3g**, 67–89%) and with excellent e.e. values (92-96%). Subsequently, we investigated α-aryl substituents on 2-acyl imidazoles (**1**) and observed that all of these substrates reacted efficiently under modified reaction conditions, affording favorable yields and 90% e.e. for compounds **3h, 3i,** and **3j**. Furthermore, we focused on investigating the scope of allylic bromide derivatives **2**. A wide variety of aryl groups incorporating electron-withdrawing or electron-donating substituents were found to participate in the reaction, giving the corresponding products **3k**–**3r** in good to high yields with excellent enantioselectivities (92–96% e.e.). The utilization of naphthyl allylic bromide exhibited similar reactivity ultimately leading to the desired product in 95% yield with 96% e.e. (**3s**). Alkyl-substituted allylic bromides were also compatible with this reaction (**3t** and **3u**).

We explored the reaction scope of *cis*-selective branched coupling of various 2-acyl imidazoles (**1**) with cinnamyl phenyl carbonates (**2**) catalyzed by (*S,S*)-**5c**-Ni/(*S,S*)-**6**-Ir (Fig. 4). 2-Acyl imidazoles bearing alkyl substituents in the α-position adjacent to the carbonyl group underwent stereoselective couplings in 58-86% yield with excellent d.r.

## Table 1 | Optimization studies

| Entry | 2 | [Ni] | [Ir] | Yield (%)[a] | 3a/4a | e.e. of 3a[b] | d.r. of 4a[c] | e.e. of 4a (%)[b] |
|---|---|---|---|---|---|---|---|---|
| 1 | **2a** | Ni(OAc)$_2$/(*S,S*)-**5a** | – | 22 | >20:1 | 28 | – | – |
| 2 | **2a** | Ni(OAc)$_2$/(*S,S*)-**5b** | – | 30 | >20:1 | 60 | – | – |
| 3 | **2a** | Ni(OAc)$_2$/(*S,S*)-**5c** | – | 80 | >20:1 | 95 | – | – |
| 4 | **2a** | Ni(OAc)$_2$/(*S,S*)-**5c** | [Ir(COD)Cl]$_2$/(*S,S*)-**6** | 83 | 1:7 | 78 | 2:1 | 85 |
| 5[d] | **2b** | Ni(OAc)$_2$/(*S,S*)-**5c** | [Ir(COD)Cl]$_2$/(*S,S*)-**6** | 83 | <1:20 | – | >20:1 | >99 |
| 6[d] | **2c** | Ni(OAc)$_2$/(*S,S*)-**5c** | [Ir(COD)Cl]$_2$/(*S,S*)-**6** | 85 | <1:20 | – | >20:1 | >99 |
| 7[d] | **2c** | Ni(OAc)$_2$ | [Ir(COD)Cl]$_2$/(*S,S*)-**6** | NR | – | – | – | – |
| 8[d] | **2c** | Ni(OAc)$_2$/(*S,S*)-**5c** | [Ir(COD)Cl]$_2$ | NR | – | – | – | – |
| 9[d] | **2c** | – | [Ir(COD)Cl]$_2$/(*S,S*)-**6** | NR | – | – | – | – |
| 10[d] | **2c** | Ni(OAc)$_2$/(*S,S*)-**5c** | – | NR | – | – | – | – |

Unless otherwise specified, all the reactions were carried out by using **1a** (0.1 mmol), **2** (0.15 mmol), Ni(OAc)$_2$ (10 mol%), **5** (10 mol%), [Ir(COD)Cl]$_2$ (2 mol%), **6** (4 mol%), and Cs$_2$CO$_3$ (0.2 mmol) in dichloromethane (DCM) (2.0 mL) at 10 °C for 48 h. [a]Isolated yields after chromatography are shown. [b]Enantiomeric excesses (e.e.) were determined by HPLC analysis. [c]Determined by $^1$H-NMR spectroscopy of the crude reaction mixtures. [d]Without Cs$_2$CO$_3$, with TBD (10 mol%) at 20 °C. TBD, 1,5,7-triazabicyclo[4.4.0]dec-5-ene. NR, no reaction.

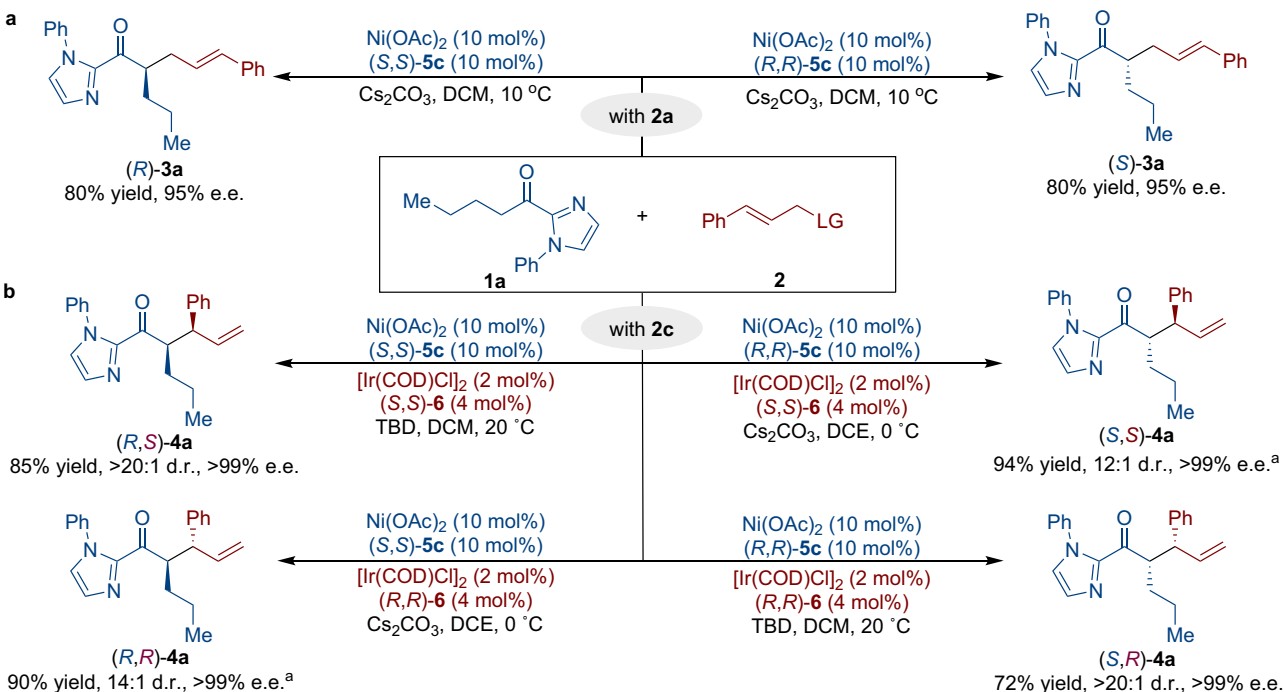

**Fig. 2 | Regio- and stereodivergent allylic alkylations. a** Asymmetric synthesis of (*S*)−**3a** and (*R*)−**3a**. **b** Stereodivergent allylations to access four stereoisomers by simply employing the enantiomer of one of the catalysts. [a]**1a** (0.3 mmol), **2c** (0.9 mmol), Cs$_2$CO$_3$ (1.0 equiv.), in DCE (1.0 mL).

values and enantioselectivities (**4a-4h**). The incorporation of a 2-phenoxy group in a 2-acyl imidazole proved to be a suitable substrate leading to excellent results (**4i**). The substrate scope of aromatic substituted acyl imidazole was investigated. Both electron-donating groups and electron-withdrawing groups proved to be compatible at the para-positon of the phenyl group, resulting in high yields and high enantioselectivities (**4j-4l**, 93-98% yield, >20:1 d.r., up to >99% e.e.). The introduction of the heterocyclic and alkyl substituents into the nitrogen atom of imidazole ring all gave the desired products (**4m-4p**) with good yields and exquisite selectivity. An X-ray crystal structure of **4n** was obtained to assign the absolute configuration of the products. The imidazole moiety was also successfully replaced by benzimidazole (**4q**), thiazole (**4r**), and pyridine (**4s**), affording the corresponding products in good yields and excellent stereoselectivities. Remarkably, the reaction exhibited notable tolerance towards both electron-donating and electron-withdrawing groups positioned differently on the phenyl ring of cinnamyl phenyl carbonates (**2**), producing compounds **4t-4aa** with enantioselectivities ranging from 98% to >99% e.e. Substrates with aromatic and heteroaromatic rings, like 2-naphthlalene and 2-thiophene undergo efficient and selective couplings (**4ab** and **4ac**). Notably, allylic carbonate containing methyl group was also tolerated (**4ad** and **4ae**).

## Synthetic application

To demonstrate synthetic utility of this methodology, a large scale reaction between **1b** and **2d** was conducted, providing the alkylation product **4af** in 83% yield with maintained stereoselectivity (Fig. 5a). Further post-functionalization of the acylimidazole group of **4af** to a series of useful synthetic building blocks is illustrated in Fig. 5a. The removal of the N-phenylimidazole moiety by the reaction with MeOTf followed with Grignard reagents gave the corresponding ketones **7** and **8** in good yields and with excellent enantiopurity. In addition, the imidazole moiety was readily removed to generate amide **9**, ester **10**, and acid **11** in high yields and with complete retention of the stereochemical information. **4af** could also be transformed to alcohol **12** and aldehyde **13** under reductive conditions with the same e.e. A tandem

deprotection/reduction amination sequence smoothly converted **4af** to amine **14**. To highlight the utility of this asymmetric allylic reaction, we applied our protocol to the synthesis of anti-Alzheimer agent arundic acid (Fig. 5b)[62, 63]. Direct alkylation of 2-acyl imidazole **1f** with allylic bromide **2a** gave **3f** in 65% yield and 94% e.e. Subsequently, a concise three-step sequence afforded (*R*)-arundic acid in 36% overall yield without loss of enantioselectivity (94% e.e.).

Our dual catalytic strategy was successful applied to the asymmetric total synthesis of cinnamomumolide, a naturally occurring and biologically active compound which is isolated from the dried tender stems of Cinnamomum cassia (Fig. 5c)[64]. The Ni((*S*,*S*)-**5c**)/Ir((*S*,*S*)-**6**) catalyzed allylation of 2-acyl imidazole **1h** with allyl aryl carbonate **2e** delivered (*S*,*S*)-**16** in 91% yield, >20:1 d.r., and 99% e.e. The subsequent synthetic sequence was initiated by a deprotection and esteration of **16** to give the corresponding ester. The subsequent sequences, which involved oxidation, reduction-cyclization, and deprotection steps, successfully yielded (*S*,*S*)-cinnamomumolide **17**[65].

The stereodivergence of the Ni/Ir dual-catalyzed allylic substitution reactions was demonstrated in the collective stereodivergent total synthesis of tapentadol[66, 67]. Under otherwise identical conditions, the enantioselective alkylation of 2-acyl imidazole **1b** with allyl aryl carbonate **2d** was carried out using four different pairs of enantiomers of the nickel catalyst and the iridium catalysts (Fig. 5d). By simple permutations of the enantiomers of the two catalysts, four stereoisomers of the desired products **4af** were generated in good yields with excellent diastereo- and enantioselectivity from the same set of starting materials. Removal of the *N*-phenylimidazole moiety of **4af** followed by amination with dimethylamine delivered the corresponding amide **18**. Through a sequential reduction of the amide and alkene functional groups, followed by a demethylation step, we achieved successful synthesis of tapentadol **19** as the final product. Thus, we have developed a collectively stereodivergent asymmetric total synthesis of tapentadol with all four stereoisomers accessible[68].

In summary, we have demonstrated the diastereodivergent allylic alkylation of 2-acylimidazoles, allowing efficient access to α-allylated tertiary carbonyls with a broad scope, high efficiency, and consistent

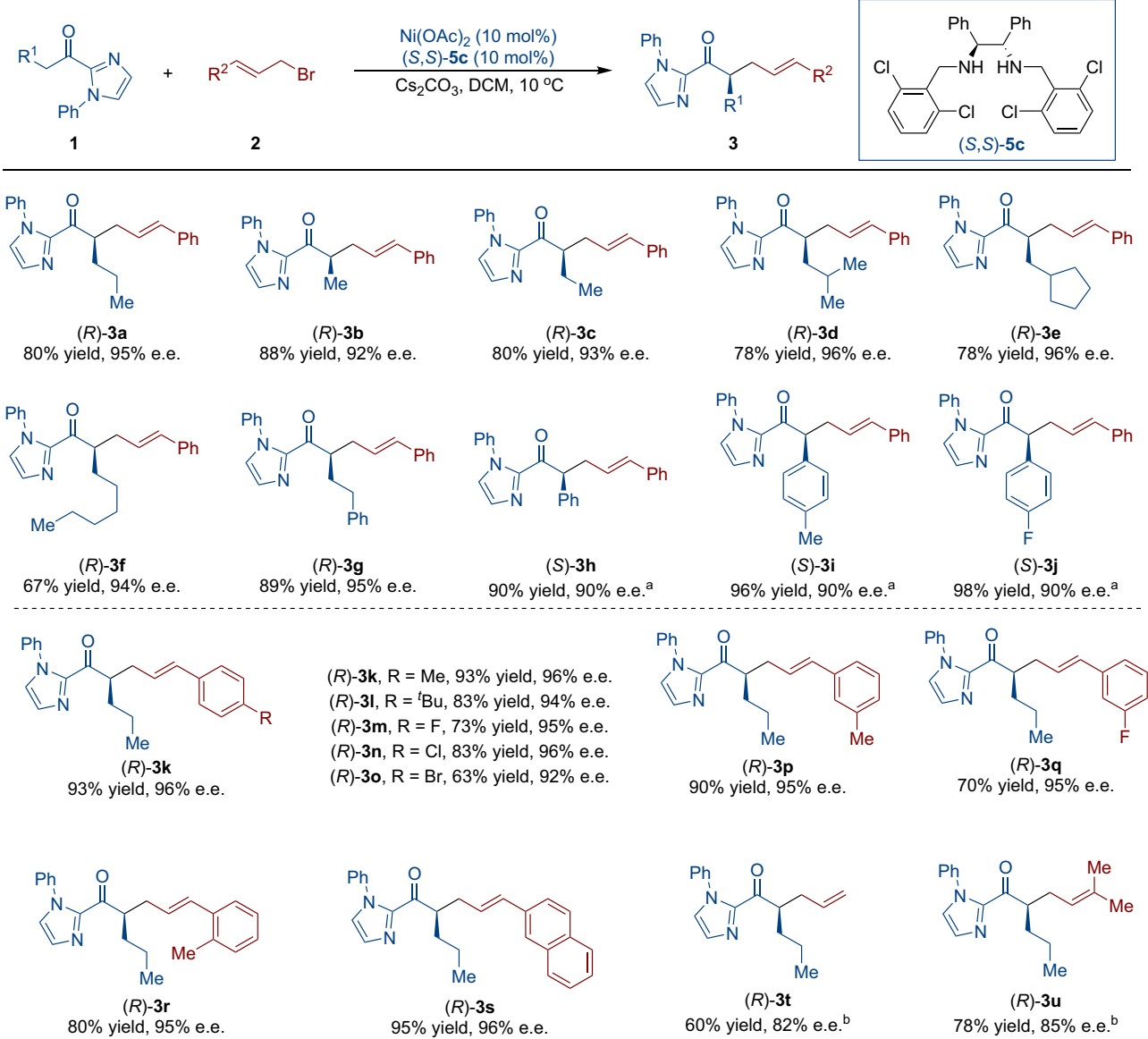

**Fig. 3 | Scope of linear allylic alkylation.** Unless otherwise specified, all the reactions were carried out by using **1** (0.1 mmol), **2** (0.15 mmol), Ni(OAc)$_2$ (10 mol%), (S,S)−**5c** (10 mol%), and Cs$_2$CO$_3$ (0.2 mmol) in dichloromethane (DCM) (2.0 mL) at 10 °C for 1-4 days. [a]With Cs$_2$CO$_3$ (0.1 mmol) at 0 °C. [b]With **2** (0.3 mmol).

stereoselectivity. This broadly applicable allylic alkylation reaction is now highly enantioselective and stereodivergent, enabling the synthesis of both linear and branched allylated products that feature privileged frameworks commonly found in biologically relevant molecules and natural products. Moreover, the innovative stereodivergent Ni/Ir dual catalysis provides a unified route to access all four stereoisomers of corresponding branched products with high diastereoselectivity and enantioselectivity by simple permutations of the enantiomers of the nickel and iridium complexes. Remarkably, the value of this operationally simple protocol has been highlighted through the collective synthesis of (R)-arundic acid and (S,S)-cinamomumulide, as well as the stereodivergent total synthesis of tapentadol.

## Methods
### General procedure A for Ni-catalyzed asymmetric allylation to generate linear product 3
To a flame-dried and argon-purged Schlenk tube were added 2-acyl imidazoles **1** (0.1 mmol, 1.0 equiv.), allylic bromide derivatives **2** (0.15 mmol, 1.5 equiv.), Ni catalyst (0.01 mmol, 0.1 equiv.), Cs$_2$CO$_3$

(0.2 mmol, 2.0 equiv.), and DCM (2 mL). The reaction mixture was stirred at 10 °C until complete consumption of the substrates (monitored by TLC). The solution was diluted with dichloromethane and then filtered with celite. The residue was purified by flash column chromatography on silica gel to afford the desired product **3**.

### General procedure B for Ni/Ir dual-catalyzed stereodivergent allylation to generate branched product 4
To a flame-dried and argon-purged Schlenk tube were added [Ir(COD)Cl]$_2$ (0.002 mmol, 0.02 equiv.), **6** (0.004 mmol, 0.04 equiv.), TBD (0.01 mmol, 0.1 equiv.), and DCM (1 mL), and the resulting solution was stirred at 25 °C for 10 min. Then, 2-acyl imidazoles **1** (0.1 mmol, 1.0 equiv.), allylic carbonates **2** (0.15 mmol, 1.5 equiv.), Ni catalyst (0.01 mmol, 0.1 equiv.), and DCM (1 mL) were added successively. The resulting solution was stirred at 20 °C. After the reaction was complete (monitored by TLC), the residue was purified by flash column chromatography on silica gel to afford the desired product **4**.

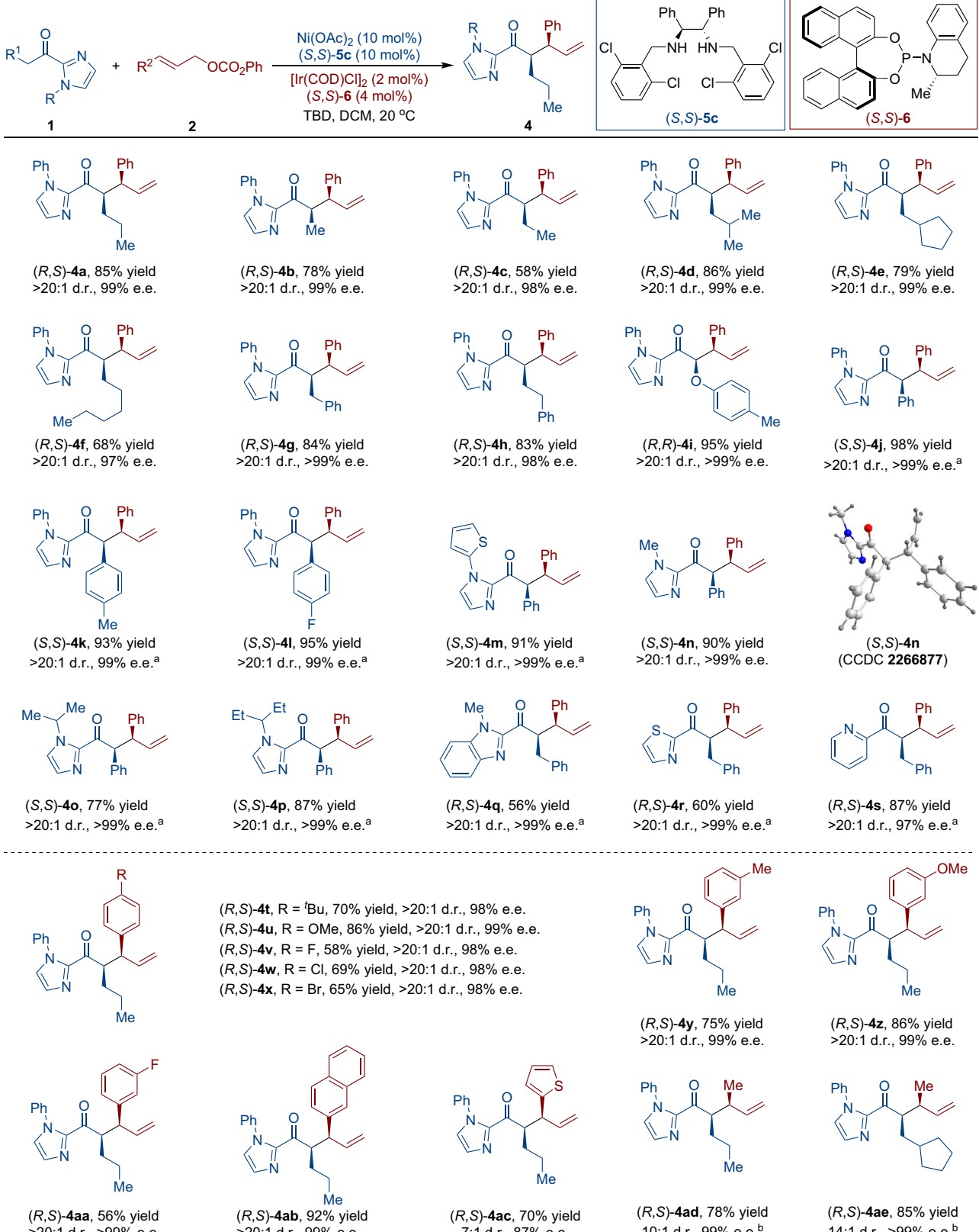

**Fig. 4 | Substrate scope of branched allylic alkylation.** Unless otherwise specified, all the reactions were carried out by using **1** (0.1 mmol), **2** (0.15 mmol), Ni(OAc)$_2$ (10 mol%), (S,S)−**5c** (10 mol%), [Ir(COD)Cl]$_2$ (2 mol%), (S,S)−**6** (4 mol%), and TBD (10 mol%) in dichloromethane (DCM) (2.0 mL) at 20 °C for 1-7 days. Branched/linear = >20:1 for all. [a]At 0 °C. [b]branched/linear = 4.6:1.

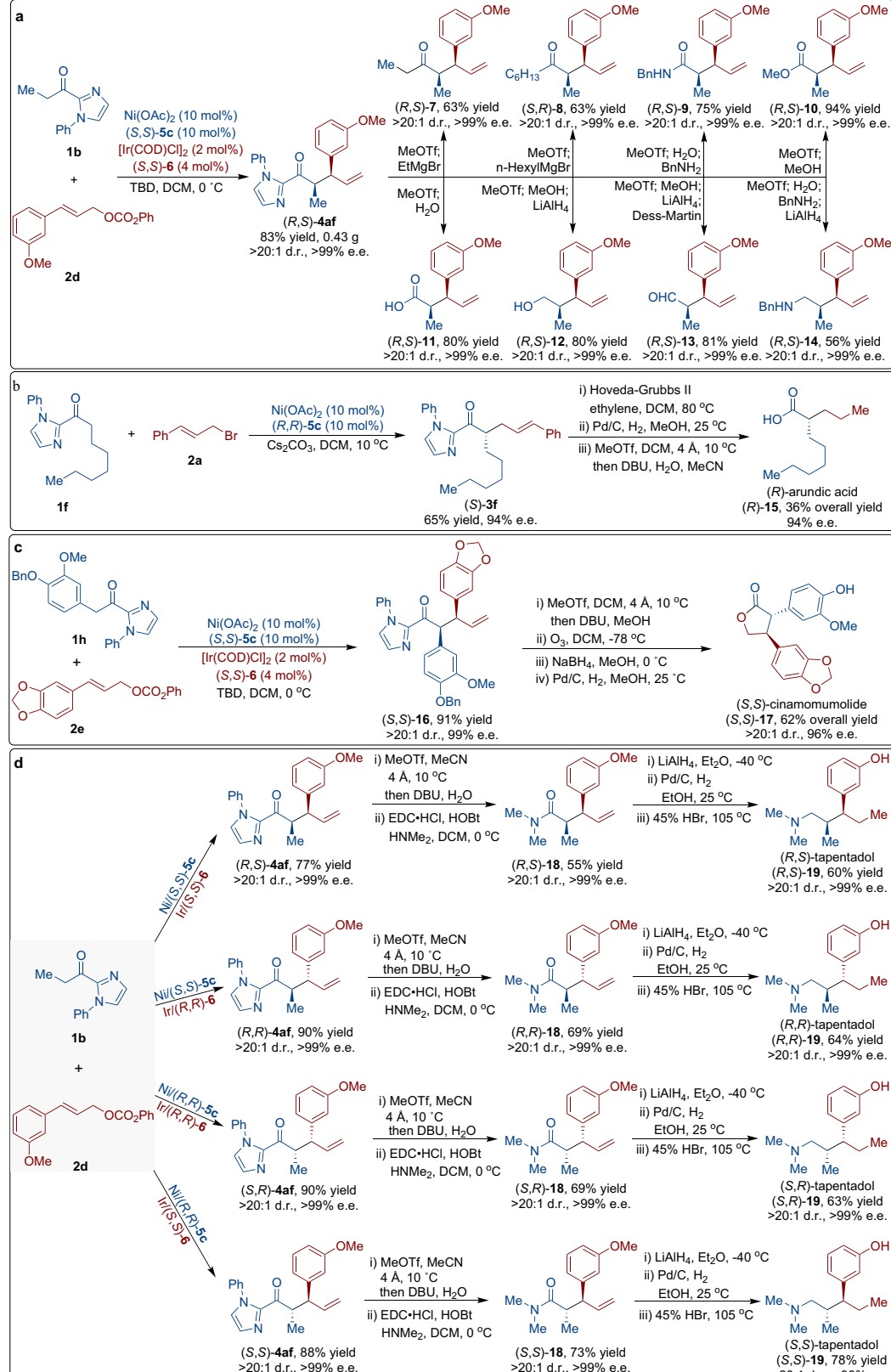

**Fig. 5 | Synthetic utility. a** Product derivation. **b** Asymmetric synthesis of (*R*)-arundic acid. **c** Asymmetric synthesis of (*S,S*)-cinamomumolide. **d** Stereodivergent total synthesis of tapentadol.

## Data availability

Crystallographic data for the structures reported in this article have been deposited at the Cambridge Crystallographic Data Centre, under deposition number CCDC 2266877 ((*S,S*)-**4n**). Copies of the data can be obtained free of charge via https://www.ccdc.cam.ac.uk/structures/. All other data supporting the findings of this study, including experimental procedures and compound characterization, NMR, and HPLC are available within the Article and its Supplementary Information or

from the authors. Source data are provided with this paper. NMR data in a mnova file format and HPLC traces are available at Zenodo at https://zenodo.org/records/10050924, under the Creative Commons Attribution 4.0 International license.

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

## Acknowledgements

The authors acknowledge financial support from the National Natural Science Foundation of China (grant no. 21971227, 22222113), CAS Project for Young Scientists in Basic Research (YSBR-054), and the Fundamental Research Funds for the Central Universities (WK9990000090, WK9990000111).

## Author contributions

C.G. conceived and designed the study and wrote the paper. R.L. and Q.Z. performed the experiments and analyzed the data. R.L. performed the stereodivergent total synthesis. All authors discussed the results and commented on the manuscript.

## Competing interests

The authors declare no competing interests.
