## [Peer Review File · Nature Communications]

REVIEWER COMMENTS

Reviewer #1 (Remarks to the Author):

This manuscript presents a novel approach for the construction of adjacent tertiary stereocenters through a synergistic allylic alkylation strategy. By employing readily diversifiable 2-acylimidazole nucleophiles and allylic electrophiles, the authors achieve impressive levels of regio-, enantio-, and diastereoselectivity across a broad range of substrates. Notably, this reaction is entirely stereodivergent, with two separate transition metal catalysts controlling each stereocenter individually. This allows for access to all possible stereoisomers of the branched alkylation products, which is highly advantageous in total synthesis applications. Furthermore, the adaptability of this asymmetric allylic alkylation was established in the enantioselective synthesis of (R)-arundic acid, (S,S)-cinamomumolide, and the stereodivergent complete synthesis of tapentadol. This study addresses the challenges associated with controlling regio-, diastereo-, and enantioselectivity in contemporary organic synthesis. Overall, the manuscript makes a valuable contribution to the field of organic synthesis by presenting a practical approach for the synthesis of chiral building blocks through stereocontrolled allylic alkylation. Given the current interest in stereodivergent total synthesis, and considering the efficacy of the dual catalytic system utilized, I do recommend publication of this manuscript in Nature Communications after minor revision.

Minor points to consider:

1. Given the alpha-stereocenter's probable sensitivity to epimerization, it would be beneficial to study whether a prolonged response time causes any erosion in diastereoselectivity.
2. In Figure 2, it would be preferable to mention the choice of different base employed for the stereodivergent allylic alkylation.
3. On page 7, line 170, "yield" should be changed to "yields."
4. Some review references on bimetallic catalysis need to be cited, which may provide useful information for the readers, for example, Chem Catalysis 2023, 3, 100455 and Chin. J. Chem. 2021, 39, 15.

Reviewer #2 (Remarks to the Author):

In this manuscript, Guo and co-workers developed a catalytic asymmetric allylic alkylation of 2-acylimidazoles to access α -allylated tertiary carbonyls with excellent outcomes. Both linear and

branched allylated products composing of six regio-, diastereo- and enantio-isomers can be readily and collectively prepared with good yields and stereoselectivities. Distinct from previous work, this approach displayed a wide range of 2-acylimidazoles bearing both alkyl- and aryl-substituents, and also achieved stereodivergent allylic alkylation through the well-accepted synergistic dual-metal catalysis. Furthermore, synthetic transformations and asymmetric synthesis of several natural products confirm its practicability and value for this protocol. Overall, this manuscript is recommended to be accepted for publication in Nature Communications after some revisions.

1) Good substrates scope has been demonstrated for both linear and branched allylic alkylation, but only aryl-substituted allylic electrophiles (-Br and -OCO₂Ph) have been reported. It is better to try or give some examples for simple alkyl-substituted electrophiles.

2) Since 2-acylimidazole contain two reactive α -C-H bonds as nucleophile and the amount of using allylic electrophiles is excess (1.5 equiv.), it is assumed whether the authors have detected any by-products, e.g. bis-allylation of 2-acylimidazole.

3) The targeted α -allylated products 3a and 4a were provided in excellent isolated yields and stereoselectivities with 10 mol% catalyst loading of chiral Lewis acid. It would be interesting to investigate the performance of low catalyst loading (e.g. 5 mol%, 1 mol%).

4) Related works on dual-metal-catalyzed allylic alkylation and application in natural product synthesis are missing. As such, Chem 2022, 8, 2011 ; Angew. Chem. Int. Ed. 2022, 61, e202115715; Angew. Chem. Int. Ed. 2022, 61, e202208837 are suggested to be cited.

5) In Page 2, line 57, there is no corresponding or malposed legend for Fig. 1e and Fig. 1f.

6) The SI is complete and reports full synthetic procedures and characterization of the allylated products. However, for readers' convenience, it is suggested to show both chemical structures and compound numbers in the NMR spectra and HPLC trace.

Reviewer #3 (Remarks to the Author):

Guo and their co-authors have developed a diastereodivergent allylic alkylation method for 2-acylimidazoles, resulting in the synthesis of α -allylated tertiary carbonyls with remarkable diastereoselectivity and enantioselectivity. This versatile allylic alkylation reaction facilitates the production of both linear and branched allylated products. The practicality of this straightforward protocol has been demonstrated through the collective synthesis of (R)-arundic acid and (S,S)-cinamomumolide, as well as the stereodivergent total synthesis of tapentadol. The manuscript is well-structured, and the writing is of good quality. The figures are presented with clarity. However, it is worth noting that while the allylic alkylation of 2-acylimidazoles leading to linear allylated products has been previously reported (ref 51), the current research employs distinct catalyst conditions and

extends the scope of application. A more comprehensive comparison between reference 51 and the present study would enhance the manuscript.

Overall, this study on the allylic alkylation of 2-acylimidazoles is suitable for publication in NC. However, there are a few points to address:

1) In Figure 1, the captions for d, e, and f appear to be disordered, both in the figure itself and within the text.

2) Regarding Table 1, it would be helpful to clarify the "-" symbol. Additionally, please provide an explanation for the differing results observed between entry 3 and entry 10.

3) In Figure 2b, the use of CsCO₃ should be explained.

4) The manuscript lacks discussion regarding reaction times, particularly in cases where the reaction proceeds slowly. Including a discussion about the reaction times would be beneficial for readers seeking to replicate the experiments.

Point-by-point response

for

Catalytic stereodivergent allylic alkylation of 2-acylimidazoles for natural product synthesis

Ruimin Lu, Qinglin Zhang, Chang Guo*

Manuscript number: NCOMMS-23-41124-T

Please find below a list of comments and changes made to the above manuscript in response to reviewers. Further changes made to the manuscript since submission are also listed at the end of this document. **A copy of the revised manuscript, a word document showing tracked changes** made to the manuscript since submission, and **revised Supplementary Information** are also included as part of this revision.

Reply to comments by Reviewer 1

1. This manuscript presents a novel approach for the construction of adjacent tertiary stereocenters through a synergistic allylic alkylation strategy. By employing readily diversifiable 2-acylimidazole nucleophiles and allylic electrophiles, the authors achieve impressive levels of regio-, enantio-, and diastereoselectivity across a broad range of substrates. Notably, this reaction is entirely stereodivergent, with two separate transition metal catalysts controlling each stereocenter individually. This allows for access to all possible stereoisomers of the branched alkylation products, which is highly advantageous in total synthesis applications. Furthermore, the adaptability of this asymmetric allylic alkylation was established in the enantioselective synthesis of (R)-arundic acid, (S,S)-cinanomumolide, and the stereodivergent complete synthesis of tapentadol. This study addresses the challenges associated with controlling regio-, diastereo-, and enantioselectivity in contemporary organic synthesis. Overall, the manuscript makes a valuable contribution to the field of organic synthesis by presenting a practical approach for the synthesis of chiral building blocks through stereocontrolled allylic alkylation. Given the current interest in stereodivergent total synthesis, and considering the efficacy of the dual catalytic system utilized, I do recommend publication of this manuscript in Nature Communications after minor revision.

Answer: We appreciate reviewer 1 for the favorable comments and helpful suggestions! These comments are greatly valuable and helpful for revising and improving our paper. We have made all the necessary amendments as suggested in our revised manuscript and revised Supplementary Information.

2. Given the alpha-stereocenter's probable sensitivity to epimerization, it would be beneficial to

study whether a prolonged response time causes any erosion in diastereoselectivity.

Answer: We thank the Reviewer 1 for the important comment. We evaluated alternative reaction times under optimum reaction conditions (Table 1, entries 3 and 6) for the allylic alkylation processes (Table S3 and Table S4). We found that extension of the reaction time with Ni/(*S,S*)-**5c** resulted in good yields with maintained e.e. (Page S26, Table S3, entries 1-3, 95% e.e.).

Table S3. Study on the results of 3a

Entry	Reaction time	Yield (%)	e.e. of 3a (%)
1	2 d	80	95
2	4 d	80	95
3	6 d	81	95

All the reactions were carried out by using **1a** (0.1 mmol), **2a** (0.15 mmol), Ni(OAc)₂ (10 mol%), (*S,S*)-**5c** (10 mol%), and Cs₂CO₃ (0.2 mmol) in dichloromethane (DCM) (2.0 mL) at 10 °C.

Meanwhile, we explored the effect of varying reaction time on the reaction outcomes for the synergistic Ni/Ir-catalytic system. We found that increasing the reaction time had no effect on the outcome (Page S26, Table S4, entries 1-3). This finding shows that the reaction system has no effect on the α -stereocenters in the products.

Table S4. Study on the results of 4a

Entry	Reaction time	Yield (%)	d.r. of 4a	e.e. of 4a (%)
1	2 d	85	>20:1	>99
2	4 d	86	>20:1	>99
3	6 d	86	>20:1	>99

All the reactions were carried out by using **1a** (0.1 mmol), **2c** (0.15 mmol), Ni(OAc)₂ (10 mol%), (*S,S*)-**5c** (10 mol%), [Ir(COD)Cl]₂ (2 mol%), (*S,S*)-**6** (4 mol%), and TBD (10 mol%) in dichloromethane (DCM) (2.0 mL) at 20 °C.

3. In Figure 2, it would be preferable to mention the choice of different base employed for the stereodivergent allylic alkylation.

Answer: With TBD as the base, the combination of Ni/(*S,S*)-**5c** and Ir/(*S,S*)-**6** leads to (*R,S*)-**4a** in 85% yield with >20:1 d.r. and >99% e.e. (Table 1, entry 6 and Fig. 2). However, the combination of Ni/(*S,S*)-**5c** and Ir/(*R,R*)-**6** leads to (*R,R*)-**4a** with unsatisfied results (24% yield, 6:1 d.r., and 99% e.e.).

All the reactions were carried out by using **1a** (0.1 mmol), **2c** (0.15 mmol), Ni(OAc)₂ (10 mol%), (S,S)-**5c** (10 mol%), [Ir(COD)Cl]₂ (2 mol%), **6** (4 mol%), and TBD (10 mol%) in dichloromethane (DCM) (2.0 mL) at 20 °C.

Further attempts to improve the yield and d.r. value of (R,R)-**4a** were investigated. To great delight, the reaction with Cs₂CO₃ as the base in DCE delivered (R,R)-**4a** in 90% yield, 14:1 d.r., and >99% e.e. (entry 9). We have included these results in our revised supporting information (Page S14, Table S1).

Table S1. Optimization studies of (R,R)-4a

Reaction scheme showing the synthesis of (R,R)-**4a** from **1a** and **2c** under optimized conditions: Ni(OAc)₂ (10 mol%), (S,S)-**5c** (10 mol%), [Ir(COD)Cl]₂ (2 mol%), (R,R)-**6** (4 mol%), Cs₂CO₃, DCE, 0 °C.

Entry	Conditions	Yield (%)	3a/4a	d.r. of 4a	e.e. of 4a (%)
1	TBD (0.1 equiv.), DCM, 20 °C	24	<1:20	6:1	99
2	NEt ₃ (1.0 equiv.), DCM, 20 °C	6	<1:20	1:1	92
3	KO ^t Bu (1.0 equiv.), DCM, 20 °C	20	<1:20	7:1	99
4	Cs ₂ CO ₃ (1.0 equiv.), DCM, 0 °C	22	<1:20	20:1	>99
5	Cs ₂ CO ₃ (1.0 equiv.), THF, 0 °C	<5			
6	Cs ₂ CO ₃ (1.0 equiv.), toluene, 0 °C	<5			
7	Cs ₂ CO ₃ (1.0 equiv.), 1,2-dichloropropane, 0 °C	20	<1:20	15:1	>99
8	Cs ₂ CO ₃ (1.0 equiv.), DCE, 0 °C	45	<1:20	17:1	>99
9 ^a	Cs ₂ CO ₃ (1.0 equiv.), DCE, 0 °C	90	<1:20	14:1	>99

Chemical structures of (S,S)-**5c** and (R,R)-**6** are shown in boxes.

All the reactions were carried out at specified conditions by using **1a** (0.1 mmol), **2c** (0.15 mmol), Ni(OAc)₂ (10 mol%), (S,S)-**5c** (10 mol%), [Ir(COD)Cl]₂ (2 mol%), (R,R)-**6** (4 mol%) in solvent (2.0 mL). ^a**1a** (0.3 mmol), **2c** (0.9 mmol), in DCE (1.0 mL).

Different enantiomers of the nickel and iridium catalysts were tested, and the combination of Ni/(R,R)-**5c** and Ir/(S,S)-**6** with Cs₂CO₃ as the base in DCE leads to (S,S)-**4a** in 94% yield, 12:1 d.r., and >99% e.e.

The reaction was carried out by using **1a** (0.3 mmol), **2c** (0.9 mmol), Ni(OAc)₂ (10 mol%), (R,R)-**5c** (10 mol%), [Ir(COD)Cl]₂ (2 mol%), (S,S)-**6** (4 mol%), and Cs₂CO₃ (0.3 mmol) in dichloroethane (DCE) (1.0 mL) at 0 °C.

4. On page 7, line 170, "yield" should be changed to "yields."

Answer: We have corrected the error in our revised manuscript.

5. Some review references on bimetallic catalysis need to be cited, which may provide useful information for the readers, for example, *Chem Catalysis* 2023, 3, 100455 and *Chin. J. Chem.* 2021, 39, 15.

Answer: As suggested by Reviewer 1, we have cited the related paper in our revised manuscript (Reference: 10 and 11 “10. Wei, L. & Wang, C.-J. Synergistic catalysis with azomethine ylides. *Chin. J. Chem.* **39**, 15–24 (2021). 11. Wei, L. & Wang, C.-J. Asymmetric transformations enabled by synergistic dual transition-metal catalysis. *Chem Catal.* **3**, 100455 (2023).”).

Reply to comments by Reviewer 2

1. In this manuscript, Guo and co-workers developed a catalytic asymmetric allylic alkylation of 2-acylimidazoles to access α -allylated tertiary carbonyls with excellent outcomes. Both linear and branched allylated products composing of six regio-, diastereo- and enantio-isomers can be readily and collectively prepared with good yields and stereoselectivities. Distinct from previous work, this approach displayed a wide range of 2-acylimidazoles bearing both alkyl- and aryl-substituents, and also achieved stereodivergent allylic alkylation through the well-accepted synergistic dual-metal catalysis. Furthermore, synthetic transformations and asymmetric synthesis of several natural products confirm its practicability and value for this protocol. Overall, this manuscript is recommended to be accepted for publication in *Nature Communications* after some revisions.

Answer: We appreciate reviewer 2 for the favorable comments and helpful suggestions! These comments are greatly valuable and helpful for revising and improving our paper. We have made all the necessary amendments as suggested in our revised manuscript and revised Supplementary Information.

2. Good substrates scope has been demonstrated for both linear and branched allylic alkylation, but only aryl-substituted allylic electrophiles (-Br and -OCO₂Ph) have been reported. It is better to try or give some examples for simple alkyl-substituted electrophiles.

Answer: As suggested by Reviewer 2, we tested the suitability of substrates with alkyl-substituted electrophiles for the synthesis of linear and branched α -allylated acylimidazoles. Remarkably, alkyl-substituted allylic bromides were successfully applied to the linear allylic alkylation reaction, giving the desired products in good yields and with good e.e. values (**3t** and **3u**). We have included these results in our revised manuscript (Fig. 3) and revised Supplementary Information (Page S12 and S13).

All the reactions were carried out by using **1a** (0.1 mmol), **2** (0.3 mmol), Ni(OAc)₂ (10 mol%), (S,S)-**5c** (10 mol%), and Cs₂CO₃ (0.2 mmol) in dichloromethane (DCM) (2.0 mL) at 10 °C.

Then we test alkyl-substituted allylic carbonates **2** for the Ni/Ir dual-catalyzed allylic substitution reactions. We initiated the study by investigating the reaction between 2-acyl imidazole **1a** and allyl phenyl carbonate **2o** in the presence of Ni/(S,S)-**5c** and Ir/(S,S)-**6** (Page S24, Table S2). However, no desired product (R,S)-**4ad** was observed (entry 1). Subsequently, the evaluation of the leaving group of the allylic carbonates (entries 2 and 3) led to the identification of the optimal conditions using substrate **2q**, providing (R,S)-**4ad** in 78% yield, 10:1 d.r., and 99% e.e. (entry 3).

Table S2. Optimization studies with alkyl-substituted allylic carbonates

Entry	2	LG	Yield (%)	Branched/linear	d.r. of (R,S)- 4ad	e.e. of (R,S)- 4ad (%)
1	2o	OCO ₂ Ph	NR	—	—	—
2	2p	OCO ₂ ^t Bu	79	3.8:1	11:1	99
3	2q	OCO ₂ Me	78	4.6:1	10:1	99

All the reactions were carried out by using **1a** (0.1 mmol), **2** (0.15 mmol), Ni(OAc)₂ (10 mol%), (S,S)-**5c** (10 mol%), [Ir(COD)Cl]₂ (2 mol%), (S,S)-**6** (4 mol%), and TBD (10 mol%) in dichloromethane (DCM) (2.0 mL) at 20 °C. NR = no reaction.

This method was compatible with 2-acyl imidazoles, giving the desired products in good yields and excellent enantioselectivities (**4ad** and **4ae**). We have included these results in our revised manuscript (Fig. 4) and revised Supplementary Information (Page S25).

All the reactions were carried out by using **1** (0.1 mmol), **2** (0.15 mmol), Ni(OAc)₂ (10 mol%), (S,S)-**5c** (10 mol%),

[Ir(COD)Cl]₂ (2 mol%), (*S,S*)-**6** (4 mol%), and TBD (10 mol%) in dichloromethane (DCM) (2.0 mL) at 20 °C.

3. Since 2-acylimidazole contain two reactive α -C-H bonds as nucleophile and the amount of using allylic electrophiles is excess (1.5 equiv.), it is assumed whether the authors have detected any by-products, e.g. bis-allylation of 2-acylimidazole.

Answer: We thank the reviewer 2 for the important comment regarding the formation of bis-allylation by-products during our reaction. After carefully monitoring our reaction by TLC, and NMR, we did not observe any formation of bis-allylation of 2-acylimidazole products and any other by-products with the use of excess allylic electrophiles.

4. The targeted α -allylated products **3a** and **4a** were provided in excellent isolated yields and stereoselectivities with 10 mol% catalyst loading of chiral Lewis acid. It would be interesting to investigate the performance of low catalyst loading (e.g. 5 mol%, 1 mol%).

Answer: As suggested by Reviewer 2, we have investigated the effect of catalyst loading on the linear allylic alkylation. We found that similar yields and stereoselectivities of the desired α -allylated product (*R*)-**3a** was achieved, when the catalyst loading was reduced to 5 mol% (Page S26, Table S5, entry 2 vs entry 1). However, it is worth noting that further decreasing the catalyst loading to 2 mol%, only 17% yield of the product was generated (entry 3). Furthermore, we observed that the reaction barely occurred when the catalyst loading was reduced to 1 mol%.

Table S5. Survey of catalyst loading for the linear allylic alkylation

Entry	Loading of $\text{Ni}(\text{OAc})_2$	Loading of (S,S)- 5c	Yield of (R)- 3a (%)	e.e. of (R)- 3a (%)
1	10%	10%	80	95
2	5%	5%	74	95
3	2%	2%	17	80
4	1%	1%	trace	-

All the reactions were carried out by using **1a** (0.1 mmol), **2a** (0.15 mmol), $\text{Ni}(\text{OAc})_2$ (X mol%), (*S,S*)-**5c** (X mol%), and Cs_2CO_3 (0.2 mmol) in dichloromethane (DCM) (2.0 mL) at 10 °C.

In addition, we also investigated the effect of catalyst loading on the branched allylic alkylation (Page S27, Table S6). We observed that reducing the catalyst loading to 5 mol% resulted in a significant decrease in both yield and enantioselectivity of the allylated product (*R,S*)-**4a** (entry 2). When the catalyst loading was further decreased to 2 mol% and 1 mol%, the reaction failed to occur (entry 3 and 4). We have included these results in our revised Supplementary Information (Table S6).

Table S6. Survey of catalyst loading for the branched allylic alkylation
Entry	Loading of Ni(OAc)_2	Loading of (S,S) - 5c	Yield (%)	d.r. of 4a	e.e. of 4a (%)
1	10%	10%	85	>20:1	>99
2	5%	5%	61	15:1	97
3	2%	2%	trace	—	—
4	1%	1%	trace	—	—

All the reactions were carried out by using **1a** (0.1 mmol), **2c** (0.15 mmol), Ni(OAc)_2 (X mol%), (S,S) -**5c** (X mol%), $[\text{Ir(COD)Cl}]_2$ (2 mol%), (S,S) -**6** (4 mol%), and TBD (10 mol%) in dichloromethane (DCM) (2.0 mL) at 20 °C.

5. Related works on dual-metal-catalyzed allylic alkylation and application in natural product synthesis are missing. As such, Chem 2022, 8, 2011; Angew. Chem. Int. Ed. 2022, 61, e202115715; Angew. Chem. Int. Ed. 2022, 61, e202208837 are suggested to be cited.

Answer: As suggested by Reviewer 2, we have cited the related paper in our revised manuscript (Reference: 31-33 “31. Xu, Y. et al. Stereodivergent total synthesis of rocaglaol initiated by synergistic dual-metal-catalyzed asymmetric allylation of benzo furan-3[2H]-one. *Chem* **8**, 2011–2022 (2022). 32. Wang, H., Xu, Y., Zhang, F., Liu, Y. & Feng, X. Bimetallic palladium/cobalt catalysis for enantioselective allylic C–H alkylation via a transient chiral nucleophile strategy. *Angew. Chem. Int. Ed.* **61**, e2021157 (2022). 33. Wang, W., Zhang, F., Liu, Y. & Feng, X. Diastereo- and enantioselective construction of vicinal all-carbon quaternary stereocenters via iridium/europium bimetallic catalysis. *Angew. Chem. Int. Ed.* **61**, e2022088 (2022).”).

6. In Page 2, line 57, there is no corresponding or malposed legend for Fig. 1e and Fig. 1f.

Answer: We have correct the error in our revised manuscript. We have included the caption for e in Fig. 1 “e, Ni/Ir dual-catalyzed stereodivergent α -allylation.” in our revised manuscript.

7. The SI is complete and reports full synthetic procedures and characterization of the allylated products. However, for readers’ convenience, it is suggested to show both chemical structures and compound numbers in the NMR spectra and HPLC trace.

Answer: As suggested by Reviewer 2, we have included both chemical structures and compounds numbers in the NMR spectra and HPLC trace in our revised supplementary information.

Reply to comments by Reviewer 3

1. Guo and their co-authors have developed a diastereodivergent allylic alkylation method for 2-acylimidazoles, resulting in the synthesis of α -allylated tertiary carbonyls with remarkable diastereoselectivity and enantioselectivity. This versatile allylic alkylation reaction facilitates the production of both linear and branched allylated products. The practicality of this straightforward protocol has been demonstrated through the collective synthesis of (R)-arundic acid and (S,S)-cinanomumolide, as well as the stereodivergent total synthesis of tapentadol. The manuscript is well-structured, and the writing is of good quality. The figures are presented with clarity. However, it is worth noting that while the allylic alkylation of 2-acylimidazoles leading to linear allylated products has been previously reported (ref 51), the current research employs distinct catalyst conditions and extends the scope of application. A more comprehensive comparison between reference 51 and the present study would enhance the manuscript. Overall, this study on the allylic alkylation of 2-acylimidazoles is suitable for publication in NC.

Answer: We appreciate reviewer 3 for the favorable comments and helpful suggestions! These comments are greatly valuable and helpful for revising and improving our paper. We have made all the necessary amendments as suggested in our revised manuscript and revised Supplementary Information. As suggested by Reviewer 3, we have described the related paper (reference 56 in our revised manuscript) in the introduction “Recently, Zheng, Wu and coworkers reported a seminal Ni/Pd dual-catalysis for the asymmetric alkylation of 2-acylimidazoles to generate linear allylated products⁵⁶.”.

2. In Figure 1, the captions for d, e, and f appear to be disordered, both in the figure itself and within the text.

Answer: We have corrected the error in our revised manuscript. We have included the caption for e in Fig. 1 “e, Ni/Ir dual-catalyzed stereodivergent α -allylation.” in our revised manuscript.

3. Regarding Table 1, it would be helpful to clarify the “-” symbol. Additionally, please provide an explanation for the differing results observed between entry 3 and entry 10.

Answer: As suggested by Reviewer 3, we have clarified the “-” symbol in Table 1, and used “NR” to show the experiment results with no reaction. Importantly, entry 3 and entry 10 utilized different leaving groups (Table 1). The corresponding allylation product was obtained in good yield with bromide as a good leaving group (entry 3). However, with less reactive carbonate ester as the substrate (entry 10), the reaction did not proceed in the absence of the iridium catalyst.

It is worth mentioning that there were slight differences in the reaction conditions between entry

3 and entry 10. In order to further investigate the impact of different leaving groups, we studied the effect of various leaving groups on the reaction under the same conditions (entry 3). We found that only bromine as the leaving group could lead to a successful reaction. This further proves that when using esters as reaction substrates, a cooperative Ni/Ir-catalytic system is necessary to activate the reactants separately and favor the α -allylation reactions (Table 1, entry 6).

Entry	2	Yield (%)	e.e. of 3a (%)
1	2a	80	95
2	2b	NR	—
3	2c	NR	—

All the reactions were carried out by using **1a** (0.1 mmol), **2** (0.15 mmol), Ni(OAc)₂ (10 mol%), (S,S)-**5c** (10 mol%), and Cs₂CO₃ (0.2 mmol) in dichloromethane (DCM) (2.0 mL) at 10 °C.

4. In Figure 2b, the use of Cs₂CO₃ should be explained.

Answer: With TBD as the base, the combination of Ni/(S,S)-**5c** and Ir/(S,S)-**6** leads to (R,S)-**4a** in 85% yield with >20:1 d.r. and >99% e.e. (Table 1, entry 6 and Fig. 2). However, the combination of Ni/(S,S)-**5c** and Ir/(R,R)-**6** leads to (R,R)-**4a** with unsatisfied results (24% yield, 6:1 d.r., and 99% e.e.).

All the reactions were carried out by using **1a** (0.1 mmol), **2c** (0.15 mmol), Ni(OAc)₂ (10 mol%), (S,S)-**5c** (10 mol%), [Ir(COD)Cl]₂ (2 mol%), **6** (4 mol%), and TBD (10 mol%) in dichloromethane (DCM) (2.0 mL) at 20 °C.

Further attempts to improve the yield and d.r. value of (R,R)-**4a** were investigated. To great delight, the reaction with Cs₂CO₃ as the base in DCE delivered (R,R)-**4a** in 90% yield, 14:1 d.r., and >99% e.e. (entry 9). We have included these results in our revised supporting information (Page S14, Table S1).

Table S1. Optimization studies of (R,R)-4a

Entry	Conditions	Yield (%)	3a/4a	d.r. of 4a	e.e. of 4a (%)
1	TBD (0.1 equiv.), DCM, 20 °C	24	<1:20	6:1	99
2	NEt ₃ (1.0 equiv.), DCM, 20 °C	6	<1:20	1:1	92
3	KO ^t Bu (1.0 equiv.), DCM, 20 °C	20	<1:20	7:1	99
4	Cs ₂ CO ₃ (1.0 equiv.), DCM, 0 °C	22	<1:20	20:1	>99
5	Cs ₂ CO ₃ (1.0 equiv.), THF, 0 °C	<5			
6	Cs ₂ CO ₃ (1.0 equiv.), toluene, 0 °C	<5			
7	Cs ₂ CO ₃ (1.0 equiv.), 1,2-dichloropropane, 0 °C	20	<1:20	15:1	>99
8	Cs ₂ CO ₃ (1.0 equiv.), DCE, 0 °C	45	<1:20	17:1	>99
9 ^a	Cs ₂ CO ₃ (1.0 equiv.), DCE, 0 °C	90	<1:20	14:1	>99

All the reactions were carried out at specified conditions by using **1a** (0.1 mmol), **2c** (0.15 mmol), Ni(OAc)₂ (10 mol%), (*S,S*)-**5c** (10 mol%), [Ir(COD)Cl]₂ (2 mol%), (*R,R*)-**6** (4 mol%) in solvent (2.0 mL). ^a**1a** (0.3 mmol), **2c** (0.9 mmol), in DCE (1.0 mL).

Different enantiomers of the nickel and iridium catalysts were tested, and the combination of Ni/(*R,R*)-**5c** and Ir/(*S,S*)-**6** with Cs₂CO₃ as the base in DCE leads to (*S,S*)-**4a** in 94% yield, 12:1 d.r., and >99% e.e.

The reaction was carried out by using **1a** (0.3 mmol), **2c** (0.9 mmol), Ni(OAc)₂ (10 mol%), (*R,R*)-**5c** (10 mol%), [Ir(COD)Cl]₂ (2 mol%), (*S,S*)-**6** (4 mol%), and Cs₂CO₃ (0.3 mmol) in dichloroethane (DCE) (1.0 mL) at 0 °C.

- The manuscript lacks discussion regarding reaction times, particularly in cases where the reaction proceeds slowly. Including a discussion about the reaction times would be beneficial for readers seeking to replicate the experiments.

Answer: We appreciate Reviewer 3 for the favorable comments and helpful suggestions! We have included the discussion regarding reaction times in our revised supporting information. For highly reactive aryl-substituted acyl imidazole substrates, the desired products can be obtained at a rapid reaction rate (**3h**). However, the allylation reaction proceeds slower and requires a longer reaction time for alkyl-substituted acyl imidazole substrates which are significantly less reactive (**3a**) (Page S28, Figure S1).

Table S7. Kinetic study for the linear allylic alkylation
Entry	R = Ph ^a		R = ⁿ Pr	
	Reaction time	Yield of 3h (%)	Reaction time	Yield of 3a (%)
1	1 h	28	2 h	17
2	2 h	40	4 h	35
3	3 h	53	6 h	42
4	4 h	64	8 h	47
5	8 h	90	12 h	55

All the reactions were carried out by using **1** (0.1 mmol), **2a** (0.15 mmol), Ni(OAc)₂ (10 mol%), (S,S)-**5c** (10 mol%), and Cs₂CO₃ (0.2 mmol) in dichloromethane (DCM) (2.0 mL) at 10 °C. ^aWith Cs₂CO₃ (0.1 mmol) at 0 °C.

**Figure S1. Kinetic profile for the linear allylic alkylation**

Additionally, we conducted an investigation on different substitution patterns of acyl imidazole substrates influencing the reaction rate of the Ni/Ir dual-catalytic system. Similarly, aryl-substituted acyl imidazole substrates showed higher reactivity in comparison to their alkyl-substituted counterparts. We have included these results in our revised Supplementary Information (Page S29, Figure S2).

Table S8. Kinetic study for the branched allylic alkylation
Entry	R = Ph ^a		R = ⁿ Pr	
	Reaction time	Yield of 4j (%)	Reaction time	Yield of 4a (%)
1	0.25 h	27	2 h	13
2	0.5 h	46	4 h	22
3	0.75 h	56	6 h	31
4	1 h	66	8 h	39
5	2 h	78	12 h	54
6	4 h	98		

All the reactions were carried out by using **1** (0.1 mmol), **2c** (0.15 mmol), $\text{Ni}(\text{OAc})_2$ (10 mol%), $(S,S)\text{-5c}$ (10 mol%), $[\text{Ir}(\text{COD})\text{Cl}]_2$ (2 mol%), $(S,S)\text{-6}$ (4 mol%), and TBD (10 mol%) in dichloromethane (DCM) (2.0 mL) at 20 °C. ^aAt 0 °C.

**Figure S2. Kinetic profile for the branched allylic alkylation**

REVIEWERS' COMMENTS

Reviewer #2 (Remarks to the Author):

Guo and co-workers have replied all the concerns and questions in a professional and accurate form regardless of the main text and SI. This reviewer strongly supports to accept this manuscript for publication in Nature Communications.